# Dynamic capillary assembly of colloids at interfaces with 10,000*g* accelerations

Axel Huerre [1], Marco De Corato[1] & Valeria Garbin [1]

High-rate deformation of soft matter is an emerging area central to our understanding of far-from-equilibrium phenomena during shock, fracture, and phase change. Monolayers of colloidal particles are a convenient two-dimensional model system to visualise emergent behaviours in soft matter, but previous studies have been limited to slow deformations. Here we probe and visualise the evolution of a monolayer of colloids confined at a bubble surface during high-rate deformation driven by ultrasound. We observe the emergence of a transient network of strings, and use discrete particle simulations to show that it is caused by a delicate interplay of dynamic capillarity and hydrodynamic interactions between particles oscillating at high frequency. Remarkably for a colloidal system, we find evidence of inertial effects, caused by accelerations approaching 10,000*g*. These results also suggest that extreme deformation of soft matter offers new opportunities for pattern formation and dynamic self-assembly.

[1] Department of Chemical Engineering, Imperial College London, London SW7 2AZ, UK. Correspondence and requests for materials should be addressed to V.G. (email: v.garbin@imperial.ac.uk)

Particles floating at liquid interfaces are a useful two-dimensional model to visualise the structure and deformation of condensed matter. A fascinating example dates back to L. Bragg, who used rafts of floating bubbles to illustrate grain boundaries and plastic flow in metals[1]. Colloidal particles confined at liquid interfaces have been used widely for this purpose, as the interparticle interactions can be finely tuned through electrostatics[2,3] and capillarity[4–6], giving access to a range of two-dimensional condensed phases[7–11]. Recently, colloids adsorbed on a curved interface were used to probe growth[12], freezing[13], healing[10] and mechanical resistance of colloidal crystals[14]. Beyond their use as two-dimensional model, colloid monolayers at interfaces, and interfacial soft matter in general, play an important role in natural and industrial processes[15]. For instance, the mechanical strength imparted to the interface by the colloids enables the formation of bicontinuous emulsions by arrested spinodal decomposition[16], suppression of the coffee-ring effect[17], and arrested dissolution of bubbles[18,19].

Previous studies of dynamic deformation of colloid monolayers have been limited to relatively low deformation rates[8,11,20–23] in the range $10^{-2}$–$1\,s^{-1}$. However, in realistic conditions, such as in the flow of emulsions and foams, and the evaporation of suspensions, interface deformations can occur on much shorter timescales, driving the system far from equilibrium. In bulk suspensions under flow, colloids form out-of-equilibrium structures stemming from the interplay of interparticle and hydrodynamic interactions[24]. Similarly, hydrodynamic interactions between colloids at interfaces can be expected to affect their assembly upon dynamic interface deformation. In more extreme conditions, phenomena that are usually not observed in a colloidal system can become important, for instance elastic collisions during shock propagation[25]. Furthermore, some of the phenomena observed for interface deformations of large amplitude, or at high rate, have no counterpart in three-dimensions: when compressed beyond hexagonal close packing, a monolayer of colloids at an interface can buckle out of plane[26,27] or expel colloids in the surrounding fluid[28,29]. Yet, the behaviour of interfacial soft matter under extreme deformation remains poorly understood due to the experimental challenge of simultaneously imparting high-rate deformation and visualising rearrangements of the microstructure.

Here we use acoustic excitation of particle-coated bubbles to explore the far-from-equilibrium phenomena of colloid monolayers at fluid interfaces. The particles, initially forming a cohesive disordered structure at rest, rearrange into a transient network of strings upon dynamic deformation. To explain this unexpected microstructure, we propose an interaction model including hydrodynamic interactions between particles oscillating at high frequency, and dynamic capillary interactions. In particular, we propose that the acceleration of the particles, approaching 10,000$g$, causes inertial effects leading to dynamic deformations of the interface, which result in capillary attraction. Discrete particle simulations based on this interaction model predict the observed microstructure in great detail.

## Results

### Evolution of the microstructure during high-rate deformation.
Bubbles (equilibrium radius $R_0 \approx 20$–$100\,\mu m$) were coated with a monolayer of polystyrene spheres (radius $a \approx 1$–$5\,\mu m$). Electrostatic repulsion between the particles was completely screened by addition of electrolyte (Methods). We found no evidence of additional electrostatic interactions arising from an asymmetric charge distribution on the particles at the interface[7]. The initial microstructure of the monolayer at moderate surface coverage was determined by capillary attraction, due to

nanoscale undulations of the contact line with an estimated amplitude $Q_2 \approx 50\,nm$[30,31]. This interaction is directional, as can be seen from a decomposition of the interface deformation in two-dimensional multipoles, which shows that contact line undulations result in capillary quadrupoles[30]. Isolated bubbles were driven into periodic compression–expansion by ultrasound at frequency $f = \omega/2\pi = 30$–$50\,kHz$, where $\omega$ is the angular frequency, in an acoustical-optical setup (Methods), and imaged at 300,000 frames per second. From the high-speed videos, we extracted the evolution of the bubble radius $R(t)$, the maximum oscillation amplitude $\Delta R$, and the trajectories of the particles on the surface of the bubble (Supplementary Note 1). The surface coverage $\Phi = \frac{N\pi a^2}{A}$, where $N$ is the number of particles in the region of interest, and $A$ is the surface area of the region of interest (Supplementary Fig. 1), and the equilibrium bubble radius, $R_0$, were measured from high-resolution still images. Figure 1a shows a sequence of frames during one cycle of oscillations of a 65-$\mu m$ bubble coated with 5-$\mu m$ particles at $\Phi = 0.54 \pm 0.04$. Particles initially at contact are driven apart during bubble expansion, and pushed back into contact during compression, as is clearly seen for the two particles marked in red and blue. The separation of the particles typically occurs in less than 10 $\mu s$.

A striking change in the microstructure of the monolayer is observed over a few hundreds of cycles of oscillations, as shown in Fig. 1b. Initially the particles, 2.5 $\mu m$ in radius and with surface coverage $\Phi = 0.48 \pm 0.05$, form a disordered, cohesive structure on the interface of a bubble with equilibrium radius $R_0 \approx 53\,\mu m$. The arrangement of the particles evolves towards a network of strings in a few hundreds of cycles (Supplementary Movies 1 and 2). The break-up of the initially aggregated structure is made possible by the mechanical energy input provided to the system during high-rate oscillations. The quadrupolar capillary attraction energy is $E_{\text{quad}} \propto \gamma Q_2^2 \sim 10^4$–$10^5\,k_B T$ at contact[32], where $\gamma$ is the surface tension, resulting in kinetically trapped structures. For comparison, the kinetic energy of a particle in our experimental conditions, based on the maximum velocity of the interface $\omega \Delta R$, is $E_k \sim a^3 \rho_P \omega^2 \Delta R^2 \sim 10^3$–$10^7\,k_B T$, where $\rho_P$ is the particle density.

The network of strings is a transient microstructure that relaxes after the forcing stops. Figure 1c shows frames of the initial, disordered microstructure before oscillations ($t = 0\,s$), of the string network formed after 1000 cycles of oscillations ($t = 0.03\,s$), and of the subsequent relaxation after the forcing has stopped ($t = 0.5$–$1.5\,s$). It can be seen that some of the strings break upon relaxation over 1.5 s. The two high-resolution still images in Fig. 1d show strings formed at low surface coverage, $\Phi \approx 0.37$ and $\Phi \approx 0.20$. The two fluorescence images (Methods) in Fig. 1d highlight additional characteristic features of the observed microstructures, such as loops ($\Phi \approx 0.30$) and lone particles ($\Phi \approx 0.43$). Loops and lone particles, neither of which are observed at rest, also disappear upon relaxation.

### Quantitative characterisation of the microstructure.
A quantitative characterisation of the microstructure shows that the formation of strings is accompanied by a decrease in the number of nearest neighbours, $n$, as can be seen by comparing the initial (Fig. 2a) and final (Fig. 2b) states of the experiment of Fig. 1b. In Fig. 2a, b, the particles are colour-coded according to the value of $n$. Initially, particles mainly have 3–5 neighbours. After 1000 cycles of oscillations, particles predominantly have 2 or 3 neighbours. The probability $p(n)$ of a particle having a number $n$ of neighbours is shown in Fig. 2c for the initial state, and in Fig. 2d for the final state. The time evolution of $p(n)$ over 1,000 cycles is shown in Supplementary Fig. 2. The statistics show clearly that particles can have up to 5 or 6 neighbours in the

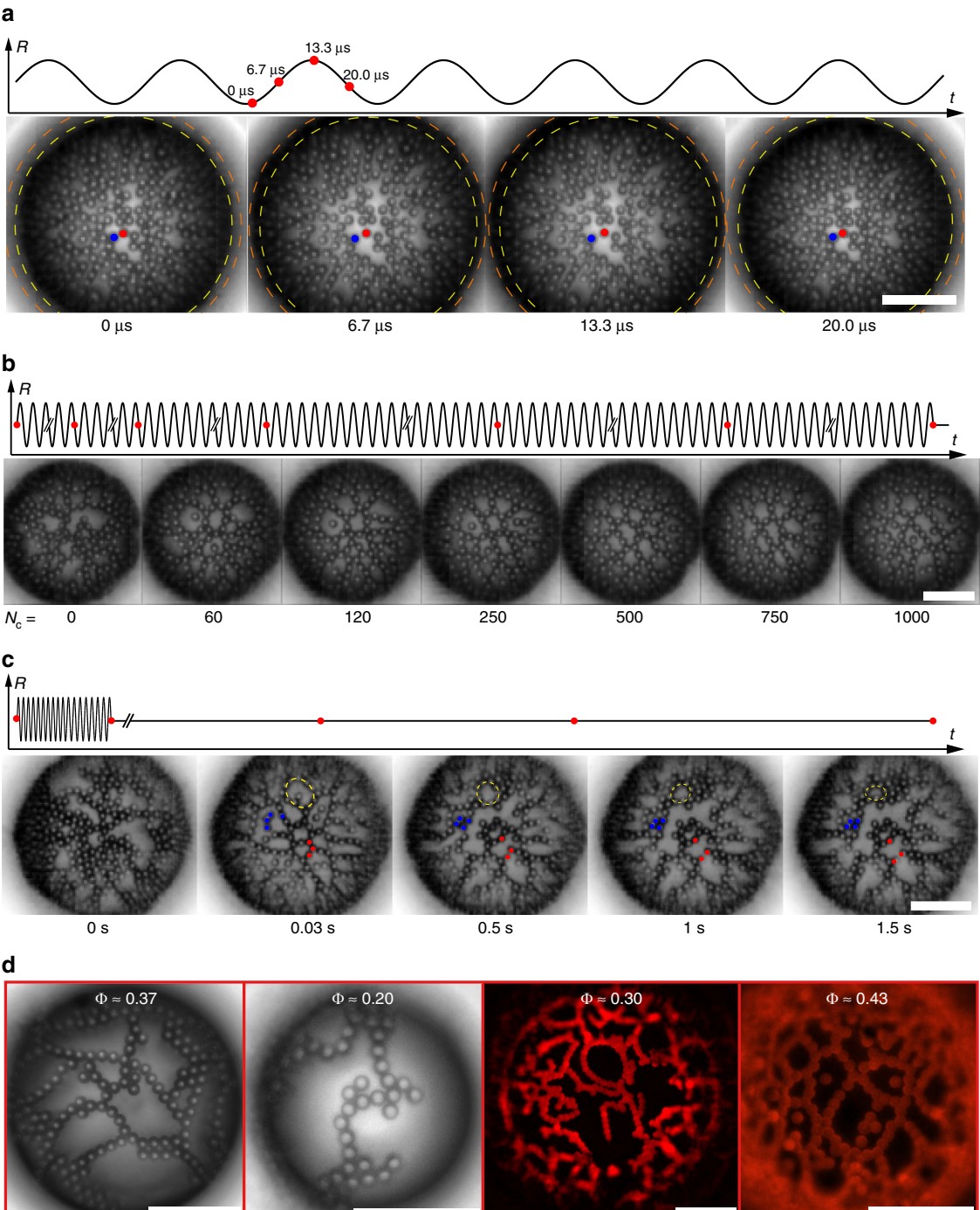

**Fig. 1** Formation of transient microstructure during dynamic deformation of the interface. **a** Time-resolved image sequence of bubble oscillations at 40 kHz ($a = 2.5\,\mu m$, $\Phi = 0.54$, $R_0 = 65\,\mu m$, $\Delta R = 6\,\mu m$). The colloids marked in red and blue move apart and come back into contact during one oscillation cycle (25 μs). The dashed circles mark the minimum (yellow) and maximum (orange) bubble radius during oscillation. **b** Image sequence of the evolution of the microstructure over 1000 cycles of oscillations ($a = 2.5\,\mu m$, $\Phi = 0.48$, $R_0 = 53\,\mu m$, $\Delta R = 1.9\,\mu m$). **c** Initial aggregated microstructure (0 s), network of strings after 1000 cycles of oscillations (0.03 s) and relaxation of the microstructure after the forcing stops (0.5–1.5 s). Some rearrangements are highlighted by the dashed lines and the colour-coded particles ($a = 2.5\,\mu m$, $\Phi = 0.46$, $R_0 = 61\,\mu m$, $\Delta R = 3\,\mu m$) **d** High-resolution bright-field and fluorescence images of networks of strings for different surface coverages. From left to right: $\Phi \approx 0.37$ (with $a = 2.5\,\mu m$, $R_0 = 59\,\mu m$); $\Phi \approx 0.2$ ($a = 2.5\,\mu m$, $R = 37\,\mu m$); $\Phi \approx 0.3$ ($a = 2\,\mu m$, $R_0 = 86\,\mu m$); $\Phi \approx 0.43$ ($a = 2\,\mu m$, $R_0 = 56\,\mu m$). Scale bars: 40 μm

initial state, whereas in the final state $p(n = 5)$ and $p(n = 6)$ become zero, and there is a sharp increase in $p(n = 2)$. Correspondingly, the mean number of neighbours per particle decreases from $\bar{n} = 3.4$ to $\bar{n} = 2.4$. Not only the number of neighbours is reduced, but the neighbours are aligned, as indicated by the increase of the bond order parameter $|\Psi_2|$ over

time, as shown in Fig. 2e (see Supplementary Note 1 and Supplementary Fig. 3). Similarly to the $|\Psi_6|$ two-dimensional bond-orientational order parameter that measures the orientation and degree of hexagonal order around a particle (6-fold symmetry), the $|\Psi_2|$ function is a metric for the alignment of particles (2-fold symmetry). We also observe an increase of the $|\Psi_3|$

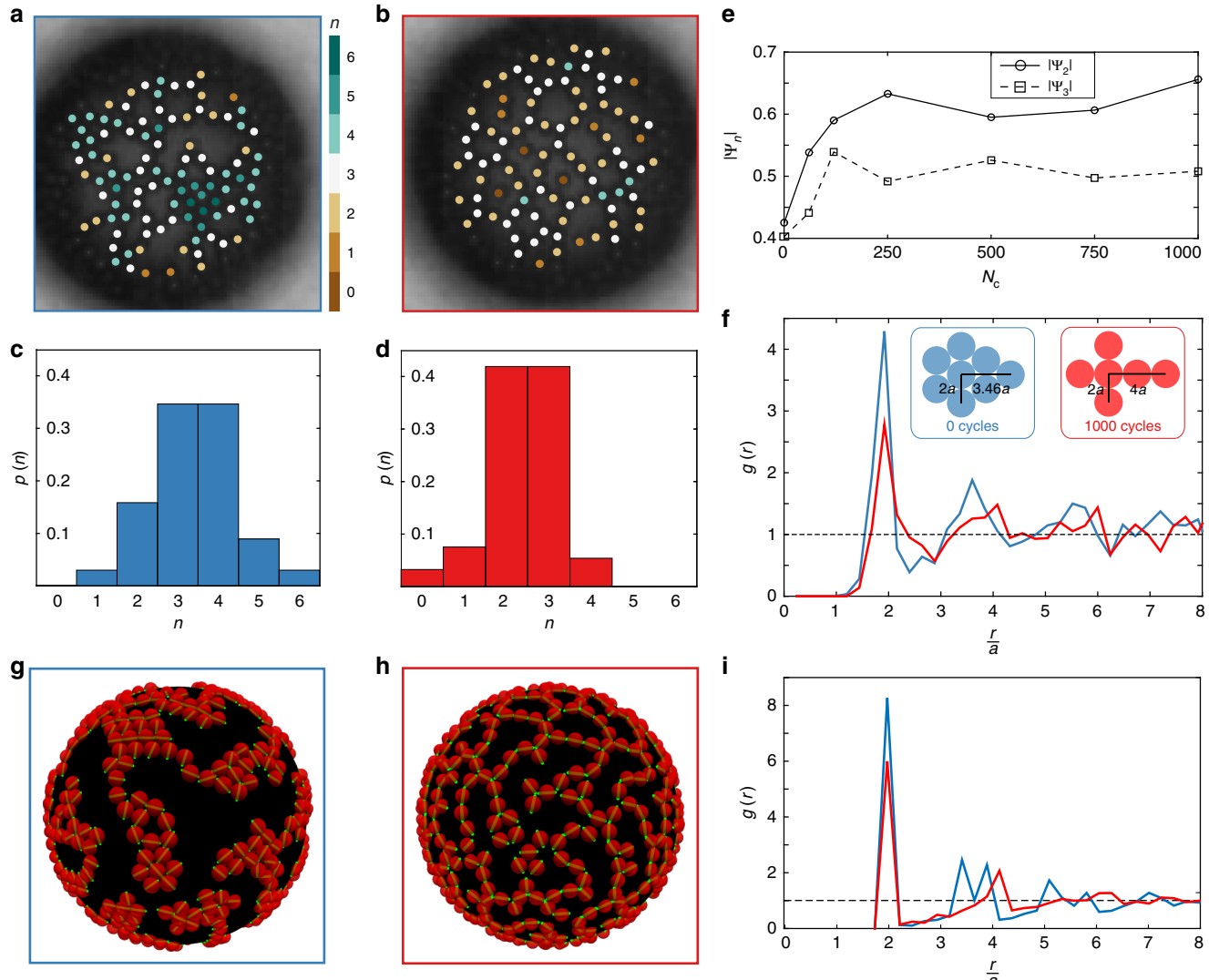

**Fig. 2** Characterisation of the microstructure and comparison with the model. **a**, **b** Images of the initial (**a**) and final (**b**) states of the experiment in Fig. 1b, with colour-coding of the particles according to the number of neighbours $n$. The emergence of a network of strings is visually apparent. **c**, **d** Probability $p(n)$ of a particle having $n$ neighbours in the initial (**c**) and final (**d**) state. In the final state, $p(n)$ becomes zero for $n = 5, 6$ and the mean number of neighbours decreases from $\bar{n} = 3.4$ to $\bar{n} = 2.4$. **e** Evolution of the order parameters $|\Psi_2|$ and $|\Psi_3|$ as a function of the number of cycles $N_c$. The evolution of the microstructure into a network of strings occurs in ~200 cycles. **f** Pair-correlation function $g(r)$ as a function of the normalised distance $r/a$ for the initial (blue) and final (red) configurations in experiment. The schematics explain the shift of the second peak upon formation of strings. **g** Simulation pictures of the initial state ($\Phi = 0.4$). Green lines show the orientation of the quadrupolar deformation. **h** Simulation pictures of the final state. The structure obtained is strikingly similar to the experimental one. **i** Pair-correlation function $g(r)$ as a function of the normalised distance $r/a$ obtained from the initial (blue) and final (red) structures in the simulations. As observed in the experiments, in the final state $g(r)$ has a peak around $r/a = 4$, typical of a string network

order parameter, representing particles having three neighbours organised with a sp² (3-fold) symmetry. The final structure is therefore formed by particles aligned in strings that are connected respecting a sp² symmetry. The evolution of the peaks in the pair-correlation function, $g(r)$, shows that the nearest neighbours remain at contact ($r = 2a$) while the second neighbours move from $r = 2\sqrt{3}a$, corresponding to hexagonal close packing, to $r = 4a$, corresponding to a chain (Fig. 2f). All the order parameters considered relax to their initial values after the forcing stops, confirming quantitatively the transient nature of the microstructure (Fig. 3 and Supplementary Fig. 4).

**Dynamic capillary interactions**. We propose that the directional interparticle force leading to the transient formation of strings is due to dynamic capillarity. Hydrodynamic interactions alone, responsible for string formation in the bulk[24], are not sufficient, because they are repulsive for oscillating particles at a fluid interface[33]. Capillary interactions with dipolar symmetry would be sufficient to drive the formation of strings at fluid interfaces[34], but they can be ruled out in our system. Indeed, although dynamic capillary dipoles can be induced by the motion of particles along an interface between two fluids with a large viscosity mismatch[35,36], in our experiments the viscous stresses due to the lateral motion of the particles are negligible compared to surface tension forces. Another possibility for a directional interaction with dipole-like symmetry is that between a monopole and a quadrupole[37] (Supplementary Fig. 5). The possibility of a monopolar deformation of the interface in our system is not immediately apparent, because it would only be expected if a body force acts on the particles[38]. In our experiments, the only

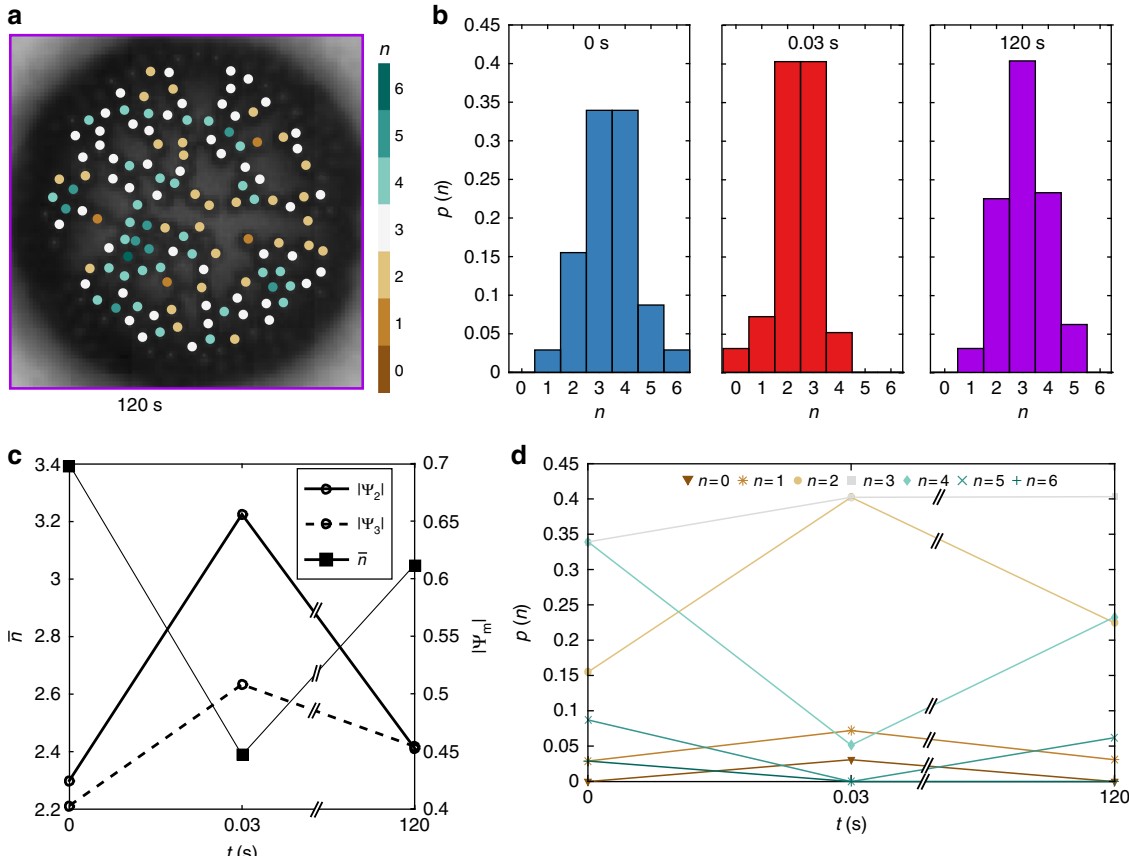

**Fig. 3** Relaxation of the microstructure after the forcing stops. **a** Image of the relaxed state 120 s after the forcing stops, for the same experiment of Fig. 2. The particles are colour-coded according to the number of neighbours $n$. **b** Comparison of the probability $p(n)$ in the initial state ($t = 0$ s, Fig. 2a); after forcing for 1000 cycles ($t = 0.03$ s, Fig. 2b); and during the relaxation process ($t = 120$ s, Fig. 3a). **c** Evolution of the mean number of neighbours $\bar{n}$ and of the bond order parameters $|\Psi_2|$ and $|\Psi_3|$ from the initial state ($t = 0$ s), to after forcing ($t = 0.03$ s), and during the relaxation process ($t = 120$ s). **d** Evolution of the probability $p(n)$ of having a number of neighbours equal to $n$ from the initial state ($t = 0$ s), to after forcing ($t = 0.03$ s), and during the relaxation process ($t = 120$ s)

body force acting on the particles is gravity, but the Bond number is in the range $Bo = \frac{\Delta \rho g a^2}{\gamma} \approx 10^{-7}$–$10^{-5}$, hence surface tension forces dominate, and the interface deformation is negligible. We hypothesise that a dynamic monopolar deformation is generated by the motion of the particle relative to the interface during bubble oscillations (Fig. 4a). The maximum acceleration of the interface is on the order of $\ddot{R} \sim \omega^2 \Delta R \approx 6000g$, an extremely large value for a colloidal system, leading to unexpected inertial effects. The Weber number based on the maximum acceleration, comparing the inertia of the particle to capillary forces, is $We = \frac{\Delta \rho \omega^2 \Delta R a^2}{\gamma} \approx 10^{-3}$–$10^{-1}$ sufficiently large for the resulting interface deformation to drive capillary interactions.

Discrete particle simulations confirm that dynamic capillary interactions between monopole and quadrupole can cause the transient microstructure observed in the experiments. The dynamics of spherical particles confined to the surface of a sphere with time-dependent radius are computed from a force balance on each particle including a simplified model for the capillary and hydrodynamic interactions. The particles are assigned a permanent quadrupolar deformation of amplitude $Q_2$, with an associated orientation vector. The inertia of a particle is assumed to cause a time-dependent monopolar deformation of the interface, $Q(t) = Q_0 \sin(\omega t)$, in phase with the motion of the interface (Supplementary Note 2 and Supplementary Fig. 6). Furthermore, a particle undergoing high-frequency oscillations in a fluid generates a steady recirculating flow, with velocity

proportional to $Q_0^2$[39]. We model the resulting hydrodynamic repulsion between particles at an interface[33] as the viscous drag experienced by a point particle in the streaming flow generated by a neighbouring particle (Fig. 4c). The pair-wise force is therefore assumed to scale as $F_{hyd} \approx \beta Q_0^2 f_{hyd}(d)$, where $d$ is the interparticle distance, and it is further assumed that $f_{hyd} = \eta \omega a^2 / d^2$ is the spatial dependence of the radial streaming velocity field generated by a neighbouring particle (Supplementary Fig. 7). For a particle at an interface, the numerical pre-factor $\beta$ is an unknown function of the Reynolds number, $Re = \frac{\rho a^2 \omega}{\eta}$, of the capillary number $Ca = \frac{\eta a \omega}{\gamma}$, with $\eta$ the viscosity of water, and of the contact angle of the particles. The case of a sphere in a bulk fluid corresponds to $\beta = 6\pi$, and $\beta$ could be larger or smaller for particles at an interface. The magnitude of the total interaction force along the line of centres of two particles $i$ and $j$ therefore has the form

$$F_{int}^{ij} = Q_0^2 f_{00}(d) \sin(\omega t)^2 + Q_0 Q_2 f_{02}\left(d, \varphi_i, \varphi_j\right) \sin(\omega t)$$
$$+ Q_2^2 f_{22}\left(d, \varphi_i, \varphi_j\right) - \beta Q_0^2 f_{hyd}(d). \tag{1}$$

The first term is the interaction between dynamic monopoles, with $f_{00} = \gamma/(2\pi d)$, the second term is the interaction between a dynamic monopole and a permanent quadrupole, and the third term is the interaction between permanent quadrupoles, with $f_{22} = (\gamma a^4)(48\pi/d^5) \cos(2\varphi_i + 2\varphi_j)$. The angles $\varphi_i$ and $\varphi_j$ define the orientation of the quadrupoles of particle $i$ and particle

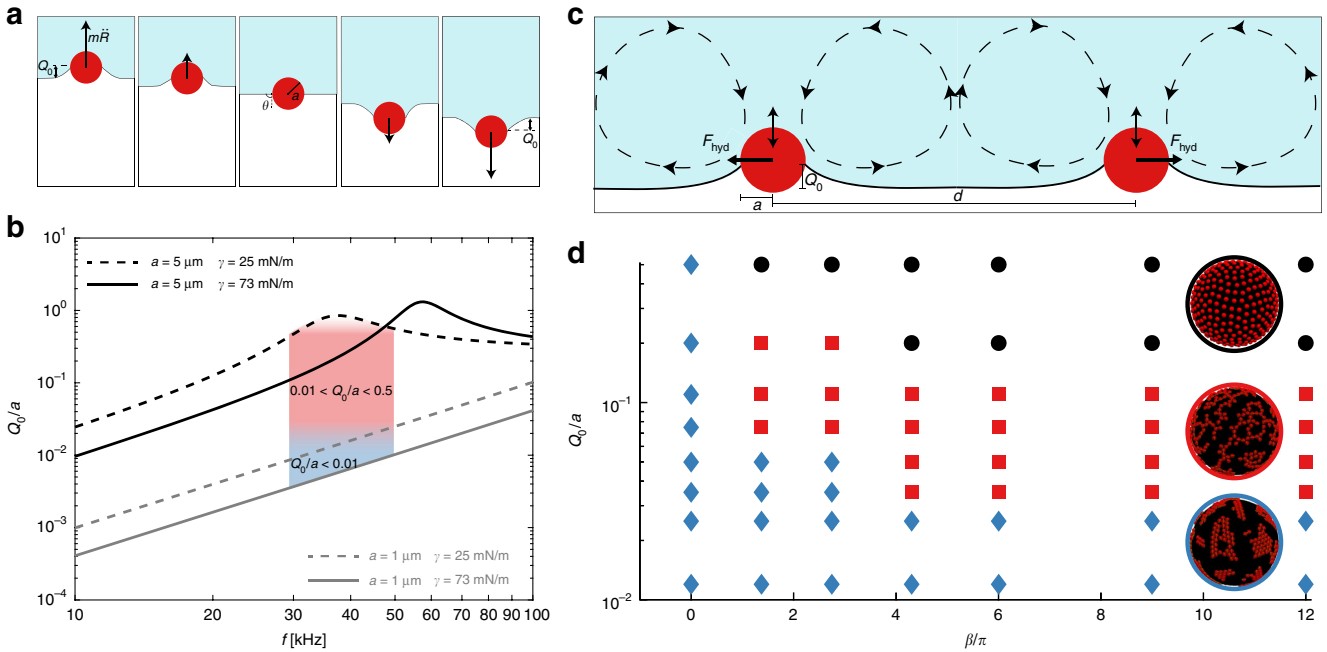

**Fig. 4** Capillary and hydrodynamic effects on particles at an oscillating interface. **a**: The interface oscillates in the normal direction with acceleration $\ddot{R}$. The inertia of the particle, $m\ddot{R}$, causes a displacement of the particle relative to the interface, resulting in a deformation of the interface of amplitude $Q_0$. **b** Amplitude of the monopolar deformation, $Q_0/a$, as a function of the frequency $f = \omega/2\pi$ obtained from a harmonic oscillator model. The particle has mass $m$ and is attached to the interface with a spring of rigidity $\gamma$ (surface tension). Four limiting cases are shown: $a = 1\,\mu m$ (grey lines) and $a = 5\,\mu m$ (black lines); $\gamma = 25\,mN/m$ (dashed lines) and $\gamma = 73\,mN/m$ (solid lines). The shaded areas correspond to small deformation (blue), and sufficiently large deformation to result in capillary interactions (red). **c** Two particles oscillating at the interface generate steady recirculating flows leading to repulsive hydrodynamic interactions. **d** Regime map of the microstructures obtained from particle based simulations as a function of the amplitude of the monopolar deformation, $Q_0/a$, and of the magnitude of hydrodynamic interactions, $\beta$. The final structures after 1000 cycles of oscillations is either an aggregated network (blue diamonds), a network of strings (red squares), or an ordered lattice (black circles), as shown in the insets

$j$ relative to the line of centres. The function $f_{02} = (\gamma a^2)(4\pi/d^3)$ $[\cos(2\varphi_i) + \cos(2\varphi_j)]$ accounts both for the effect of monopole $i$ on quadrupole $j$, and of monopole $j$ on quadrupole $i$.

We first equilibrate the system with only interactions between permanent quadrupoles ($Q_2/a = 1/100$). The resulting aggregated structure (Fig. 2g) presents rafts of particles with hexagonal packing and herringbone alignment of the orientation vectors[40]. The dynamics are simulated by allowing the radius of the sphere to vary as $R(t) = R_0 + \Delta R \sin(\omega t)$, and by including the effect of the dynamic monopole. The amplitude of the oscillatory monopolar deformation is set to $Q_0/a = 7 \times 10^{-2}$ and the other parameters are in the range of those used in the experiments (Supplementary Note 3).

The microstructure after 1000 cycles of oscillations, shown in Fig. 2h, is a network of strings with a striking similarity to the experimental results (see Supplementary Movie 5). The evolution of the pair correlation function $g(r)$ confirms quantitatively the formation of strings (Fig. 2i). In keeping with the experimental results, the first peak (at $r = 2a$) decreases in amplitude and the second peak shifts from $r = 2\sqrt{3}a$ to $r = 4a$, the signature of evolution from hexagonal close-packing to a string of particles. The orientation vectors of particles inside a string are aligned end-to-end. The evolution of $p(n)$, $\overline{n}$, $|\Psi_2|$ and $|\Psi_3|$ (Supplementary Fig. 8) are all in good agreement with the experimental measurements. The spherical shape of the interface allows to detect the effect of the dynamic monopole, which would otherwise be time-reversible on a planar interface undergoing oscillations, because the interparticle distance is not constant during one cycle of oscillations. The distance between two particles positioned on a sphere having time-dependent radius $R(t) = R_0 + \Delta R \sin(\omega t)$ varies as $d(t) = d_0 + \Delta d \sin(\omega t)$. If the

particles are initially in contact, the maximum separation distance is given by $\Delta d = a\Delta R/R_0$. The time average of the monopole–quadrupole force over one period of oscillations, $T = 2\pi/\omega$, is given by $\langle F_{02} \rangle_T \propto -\frac{3}{2}\gamma a^2 Q_0 Q_2 \frac{1}{d_0^3} \frac{\Delta d}{d_0}$ and is therefore non-zero only for a curved interface ($\Delta d \neq 0$). Simulations of the relaxation of the microstructure after the forcing stops also reproduce the behaviour observed in experiments: the order parameters recover their initial values before forcing (Supplementary Fig. 9).

**Estimate of the dynamic monopole amplitude.** To verify that the amplitude of the dynamic monopolar deformation that gives agreement with the experiments is physically justified, we use a harmonic oscillator model of a particle attached to an interface by capillary force[41] (Supplementary Note 2). The dimensionless displacement $x$ of the particle relative to its equilibrium position provides an estimate of the amplitude of the monopolar deformation, $Q_0/a$. The normalised relative displacement $Q_0/a$ as a function of the oscillation frequency $f = \omega/2\pi$ is shown in Fig. 4b for four cases, using limiting values of the mass $m$ and spring constant $\gamma$ corresponding to the experimental parameters (grey lines: $a = 1\,\mu m$; black lines: $a = 5\,\mu m$; solid lines: $\gamma = 73\,mN/m$; dashed lines: $\gamma = 25\,mN/m$). In the range of frequencies used in the experiments, highlighted by the shaded area, the monopole amplitude varies from $Q_0/a \sim 10^{-3}$, up to $Q_0/a \sim 1$. In the range $Q_0/a \sim 10^{-2}$–$10^{-1}$ (red shaded area in Fig. 4b), the monopole amplitude is sufficiently large to drive capillary interactions between particles, as has been observed in experiments with heavy particles[42]. The value $Q_0/a = 7 \times 10^{-2}$ used in the simulations of Fig. 2 is therefore justified. For amplitudes smaller than $Q_0/a \sim 10^{-2}$ (blue shaded area in Fig. 4b), the interface

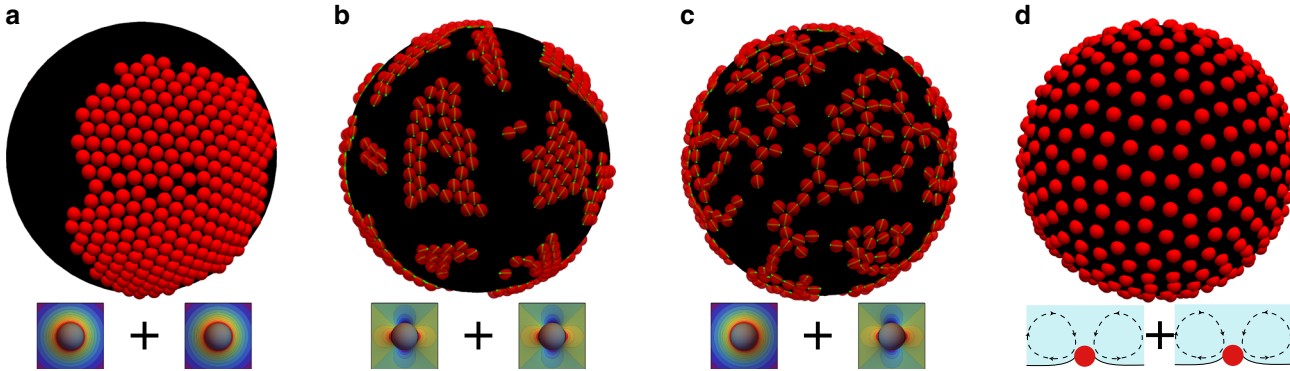

**Fig. 5** Effects of individual interactions on the microstructure. **a** Monopole–monopole interactions alone result in a large aggregate containing all the particles. **b** Quadrupole–quadrupole interactions alone result in small aggregates. **c** Monopole–quadrupole interactions alone result in the formation of strings. **d** Hydrodynamic repulsion alone results in an ordered lattice

deformation is expected to be insufficient to drive significant interactions. For $Q_0/a \sim 1$ and larger, detachment of particles from the interface can be expected[27,41].

**Conditions for string formation in simulations.** Discrete particle simulations predict string formation for physically realistic values of the monopole amplitude, $Q_0$, and of the pre-factor of the hydrodynamic interaction, $\beta$, which are the two unknown parameters in Eq. (1). We performed a parametric study for values of $\beta$ varying over more than one order of magnitude, and values of $Q_0$ varying in the range predicted by the harmonic oscillator model. The results are presented in the regime map in Fig. 4d. The microstructure obtained after 1000 cycles is classified according to the values of $p(n)$ and $g(r)$ as either an aggregated network (blue diamonds), a network of strings (red squares), or an ordered lattice (black circles). The final microstructure is found not to depend very strongly on the parameter $\beta$. However if $\beta = 0$ the structure does not evolve from an aggregated network, suggesting that repulsive hydrodynamic interactions are responsible for the break-up of the initially aggregated structure. The emergence of strings depends strongly on $Q_0/a$, which determines the magnitude of the monopole–monopole attractive force, of the directional monopole–quadrupole interaction, and of the repulsive hydrodynamic force (see Eq. (1)). An aggregated network is obtained when dynamic effects are not important ($Q_0/a \lesssim 10^{-2}$), and capillary interactions between permanent quadrupoles dominate, giving a final structure that remains similar to the initial structure. Formation of strings is predicted for $Q_0/a \sim 10^{-2}$–$10^{-1}$, which is indeed the range in which monopolar deformations are sufficiently large to cause capillary interactions[42]. When the monopole amplitude exceeds a critical value, $Q_0/a \gtrsim 10^{-1}$, the repulsive hydrodynamic interactions are found to dominate, leading to the formation of an ordered lattice. This ordered structure is never observed in the experiments, likely because it is in the range of deformations where particle expulsion occurs. Additional simulations are shown in Fig. 5 for limiting cases where only one type of interaction is included in the model. These results highlight the effect of each type of interaction on the final microstructure.

**Experimental conditions for string formation.** String formation is robust over a broad range of experimental conditions. The two control parameters for the emergence of strings are the surface coverage and the Weber number. The surface coverage determines the extent to which rearrangements of the microstructure are possible. Random close packing of spheres in two dimensions occurs at $\Phi_{rcp} \approx 0.82$, and hexagonal close packing at $\Phi_{hcp} \approx 0.91$[43]. In the experiments, the surface coverage was varied in

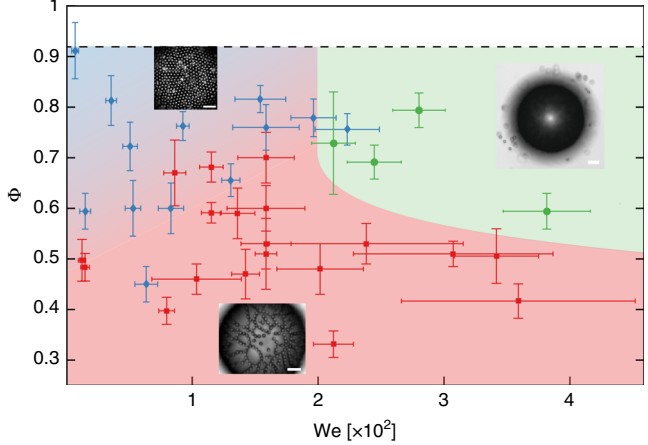

**Fig. 6** Experimental regime map for string formation. The three possible outcomes of an experiment are: formation of strings (red squares); no evolution of the microstructure (blue diamonds); or expulsion of particles (green circles). The shaded areas are a guide for the eye. The regime map shows that the outcome is controlled by the Weber number, We, comparing the inertia of a particle with surface tension forces, and the surface coverage $\Phi$. At surface coverages below $\Phi \sim 0.6$, string formation is observed for a broad range of experimental conditions. Error bars on the surface coverage are derived from extreme values of the area considered, computed considering the convex hull and the minimum circle enclosing all the particles. Error bars on We are given by $d\mathrm{We} = \mathrm{We}\left(\frac{d\Delta R}{\Delta R} + 2\frac{da}{a}\right)$. Further details are given in Supplementary Note 1. Scale bars: 40 μm

the range $\Phi \approx 0.33$–$0.90$ (Supplementary Fig. 10). The Weber number, which determines the magnitude of the dynamic monopolar deformation, was varied in the range $\mathrm{We} \approx 1 \times 10^{-3}$–$5.5 \times 10^{-2}$ by varying the particle size ($2a = 1.8, 3, 5, 10$ μm), the oscillation frequency ($f = 30, 40, 50$ kHz), and the amplitude of oscillations ($\Delta R \approx 0.2$–$3.5$ μm, with $\Delta R/R_0 \approx 0.003$–$0.13$ and $a/R_0 \sim 10^{-2}$–$10^{-1}$). We performed over 40 experiments, with three possible outcomes: no evolution of the microstructure, emergence of strings, or expulsion of particles from the interface (see Supplementary Movies 2–4 respectively). The results are presented in the ($\Phi$, We) plane in Fig. 6. The area highlighted in blue, at high $\Phi$ and low We, corresponds to experiments where the structure did not evolve during 1000 cycles of oscillations. In these conditions, rearrangements are limited due to jamming of the monolayer, and the magnitude of the dynamic interactions is insufficient to drive an evolution of the microstructure. The area highlighted in green corresponds to experiments where the amplitude of oscillations is so large that particles are expelled

from the interface[28]. In this regime it is no longer possible to observe structures at constant Φ. Finally, the area highlighted in red corresponds to string formation. Even though this microstructure results from a delicate interplay between hydrodynamics and dynamic capillary interactions, it is consistently observed for a broad range of accessible experimental conditions.

## Discussion

In summary, we have observed and explained the emergence of a transient microstructure formed by colloids undergoing extreme accelerations at a fluid interface. We have shown that dynamic interface deformation can be used to break kinetically trapped structures, and to impart dynamic capillary interactions. Despite being transient, the microstructure has a sufficiently long relaxation time that it could be exploited for subsequent processing by high-throughput methods, for instance photopolymerisation[44]. Our results therefore present new opportunities for programmable self-assembly and pattern formation in soft matter at interfaces. Beyond these potential applications, our experimental approach opens the way to future studies of interfacial soft matter far from equilibrium, at deformation rates that are otherwise inaccessible to existing techniques, but relevant to realistic flow and deformation conditions.

## Methods

**Particle-coated bubbles.** Particle-coated bubbles were made by mechanical agitation of an aqueous suspension containing 0.4% w/v of colloids, using a vortex mixer. We used hydrophilic, spherical microparticles of nominal diameter 1.8, 3, 5 and 10 μm with sulfate surface groups (IDC surfactant-free latex particles, Life Technologies). For fluorescence imaging we used red fluorescent microparticles, 4 μm in diameter, sulfate coated (FluoSpheres, Invitrogen). All particles were used as received. Addition of NaCl (VWR Chemicals, AnalaR NORMAPUR, 99.5%) with 500 mM concentration was necessary to promote particle adsorption to the water–air interface. The resulting Debye length is 0.5 nm. Ultrapure water with resistivity 18.2 MΩ cm was produced by a Milli-Q filtration system (Millipore).

**Acoustical-optical setup.** We injected particle-coated bubbles in an observation cell made of a microscope slide and a glass coverslip separated by a 1-mm spacer. Prior to every experiment, the cell was rinsed with ethanol and ultrapure water, and dried with compressed air. Small numbers of bubbles were injected in the chamber, and we only observed isolated bubbles (at least 10 diameters away from other bubbles). The observation chamber was placed on an inverted microscope (IX71, Olympus) equipped with 10× and 20× objectives. A single-element piezoelectric transducer (SMD50T21F45R, Steminc) with resonance frequency (45 ± 3) kHz was glued to the glass slide. The driving signal was generated by a waveform generator (33220A, Agilent) and amplified by a linear, radio-frequency power amplifier (AG1021, T& C Power Conversion Inc.). The frequencies used were 30, 40 and 50 kHz, resulting in different bubble oscillation amplitudes. Since the wavelength of ultrasound at these frequencies in water is λ > 3 cm, the pressure can be considered to be uniform over distances of the order of the bubble size. The bubbles were driven for 1000 cycles and the dynamics recorded at 300,000 frames per second using a high speed camera (FASTCAM SA5, Photron). The waveform generator and the high-speed camera are triggered simultaneously using a pulse-delay generator (9200 Sapphire, Quantum Composer). The image resolution at 10× and 20× magnification is 2 and 1 μm, respectively. High-resolution still images in different focal planes were taken before and after excitation at 32× magnification using a CCD camera (QImaging), resulting in an image resolution of 0.1 μm. For fluorescence imaging, illumination was provided by a Lumen 200 (Prior Scientific) UV lamp combined with a ET mCH-TR (Chroma) fluorescence cube.

**Interaction model and discrete particle simulations.** We consider $N$ spherical particles confined to a spherical surface with a time-dependent radius. The particles are assumed to interact through capillary forces and torques induced by static and dynamic deformations of the spherical surface, and through hydrodynamic forces. Brownian motion is neglected as the total duration of the experiments is much smaller than the diffusion timescale of the colloids. The irregular contact line of each particle generates a static quadrupolar deformation of the spherical surface[30], which is characterised by an orientation vector $\mathbf{p}_i$ (Supplementary Fig. 11). The overlap of the deformations induced by two quadrupoles drives an interaction force $\mathbf{F}_{22}^{*ij}$ and a torque $\mathbf{T}_{22}^{*ij}$ between particle $i$ and particle $j$, which depends on the relative orientation of the quadrupoles $\mathbf{p}_i$ and $\mathbf{p}_j$[38]. In addition, we assume that each colloidal particle imparts a periodic monopolar deformation of the interface, which is responsible for additional dynamic capillary interactions. The overlap of two dynamic monopolar deformations generated by particle $i$ and $j$ drives an attractive

capillary force $\mathbf{F}_{00}^{*ij}$[38]. The superposition of a monopolar and a quadrupolar deformation drives two capillary interaction forces $\mathbf{F}_{02}^{*ij}$ and $\mathbf{F}_{20}^{*ij}$ and a capillary torque $\mathbf{T}_{20}^{*ij}$, which depend on the orientation of the quadrupole. Finally, the relative motion between a particle and the interface generates a net recirculating flow[39] (see Fig. 4c) that is responsible for an hydrodynamic repulsive force $\mathbf{F}_{hyd}^{*ij}$. The detailed expression of the interaction forces and torques is reported in Supplementary Note 3. The evolution of the position of each particle $\mathbf{r}_i^*$ and of the orientation of the quadrupole $\mathbf{p}_i$ are obtained solving the dimensionless balances of linear and angular momentum:

$$\frac{4\pi}{3}\mathrm{St}\frac{d\mathbf{v}_i^*}{dt^*} = -3\pi\mathbf{v}_i^* + \mathbf{F}_S^{*i} + \sum_{j\neq i}\left(\mathbf{F}_{00}^{*ij} + \mathbf{F}_{22}^{*ij} + \mathbf{F}_{02}^{*ij} + \mathbf{F}_{20}^{*ij} + \mathbf{F}_{EV}^{*ij} + \mathbf{F}_{hyd}^{*ij}\right), \quad (2)$$

$$\frac{d\mathbf{r}_i^*}{dt^*} = \mathbf{v}_i^*, \quad (3)$$

$$\frac{8\pi}{15}\mathrm{St}\frac{d\boldsymbol{\omega}_i^*}{dt^*} = -4\pi\boldsymbol{\omega}_i^* + \sum_{j\neq i}\left(\mathbf{T}_{22}^{*ij} + \mathbf{T}_{20}^{*ij}\right), \quad (4)$$

$$\frac{d\mathbf{p}_i}{dt^*} = \boldsymbol{\omega}_i^* \times \mathbf{p}_i, \quad (5)$$

where $\mathrm{St} = \rho_p a^2 \omega / \eta$ is the Stokes number comparing the relative importance of the inertia of a particle and the viscous force acting on it. In typical experimental conditions $\mathrm{St} \approx 1$. Since the particles are only partially immersed in water, we assumed that the drag coefficient in Eqs. (2)–(4) is equal to half of the Stokes drag. To confine the particles to the surface of the sphere, we included a stiff restoring force $\mathbf{F}_S^{*i}$ in the balance of linear momentum. The overlap between particle $i$ and particle $j$ is prevented by enforcing a stiff excluded volume force $\mathbf{F}_{EV}^{*ij}$. Eqs. (2)–(5) are solved using a second-order explicit linear multistep method. We used a dimensionless time step $\Delta t^* = 3 \times 10^{-3}$, which has been checked to give convergent results. To avoid expensive computations we enforce a cut-off length of $7a$ for all the interaction forces and torques between the particles. Simulations with larger and smaller cut-off lengths were found to give similar results. The computational time of evaluating the interactions between the particles scales as $N^2$. The computation of the interactions was therefore parallelised over ten or more cores.

## Data availability

The experimental data that support the findings of this study and the computer code used to generate the numerical results are available from the corresponding author upon reasonable request.

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

## Acknowledgements

We thank L. Botto and A. Striolo for useful discussions. This work is supported by the European Research Council, Starting Grant No. 639221 (ExtreFlow).

## Author contributions

A.H. and V.G. designed the experiments. A.H. performed the experiments and analysed the data. M.D.C. designed and implemented the simulations. A.H., M.D.C. and V.G. interpreted the results and wrote the manuscript. V.G. designed the research.

## Additional information

**Competing interests:** The authors declare no competing interests.

