## [Peer Review File · Nature Communications]

Reviewers' comments:

Reviewer #1 (Remarks to the Author):

The authors of this paper report fantastic experiments on the dynamical assembly of microparticles trapped at the air-water interface, i.e. on the surface of a bubble, and driven by high-frequency compression-expansion cycles. Driven by inertia, the particles, which spontaneously possess static capillary quadrupoles due to contact line pinning along an undulated line, take up additional interactions during the oscillations, i.e. capillary monopoles driven the inertial deformation of the interface and repulsive hydrodynamic forces coming from recirculation flow. The delicate balance of these dynamic interactions affects the microstructure of the interface leading to the transient formation of networks of string-like aggregates, which relax and disappear after cessation of the oscillations.

The experiments are highly challenging, given the extreme deformation rates, and open up a new range of phenomena that are usually inaccessible for colloidal systems. The interpretation of the data, aided by careful numerical simulations, is extremely elegant and supported by extensive sets of controls. The paper is very well written and easily accessible for the general readership of Nature Communications. I can only recommend publication.

Before that, I have a small set of remarks and questions, that I would like to point out to the authors.

1) In the videos capturing the full dynamics during the cycles, e.g. in particular Video S1, it is clearly visible that only some of the initial aggregates are rearranged and that the interparticle distance between different particles oscillates by different extents. Can the authors comment on this issue? Is it to be understood as only coming from polydispersity of the quadrupolar interactions between particle pairs? Or is there the possibility that other elements partake the force balance? In particular, I am thinking about higher-order capillary interactions, where possible short-range repulsion can exist (e.g. as shown in the previous works of one of the authors *Soft Matter*, 2013,9, 779-786). The exploration of these effects, both polydispersity and additional interactions is highly interesting, and worth discussing.

2) The authors mention in passing (page 8) the role of the curvature of the interface, emphasizing that the effect of the dynamic monopoles is observable thanks to the spherical curvature and that a flat interface would not give the same effect. It would be useful for the reader to have more details on this, especially in the sense of relative inter-particle separation to R_0 , to identify when can the interface be assumed as flat, e.g. to guide new experiments. It is not entirely clear to me where the role of the interface curvature directly enters the various force components in Eq. (1).

3) Can the authors say something on the relaxation of the local direction of the quadrupoles of each particle after cessation of the oscillations? Do the particles suddenly reorient in local clusters with minimal interface deformation or is there a gradual re-orientation? This can only be accessed in the simulations, but it is interesting as a question for the design of future experiments.

4) I was intrigued by the reference to shock propagation in colloidal systems, but the reference appears to be incorrect, since this work covers experiments on slow oscillatory shear.

Reviewer #2 (Remarks to the Author):

This manuscript presents some unique experiments on particles on bubble interfaces and suggest a subtle interplay between capillarity and hydrodynamic disturbance velocity fields, which is termed "dynamic capillarity" by the authors. The authors study pattern formation as a function of frequency and composition and compare experiments at high frequencies with simulations.

The authors demonstrate experimentally the occurrence of intriguing stringlike structures, at low to intermediate volume fractions, which naively would bring up the idea of a dominant dipolar interaction. This brings me to a first question: - whereas the authors rule out all electrostatic effects - due to screening by salt in the water phase - what happens on the air side? Static friction effects have been suggested to play a role in oil water emulsions (see Scientific Reports volume 4, Article number: 4778 (2014)) - is there a possible residual charge or a static charging by friction with the air? I'm personally skeptical of the relevance of the charging reported in scientific reports, but it may be something to consider or at least comment on.

The authors argue that another possibility for a directional interaction with dipole-like symmetry is that between a monopole and a quadrupole. The hypothesis that capillary deformations forces due to contact line undulations interact with dynamic monopoles seems to be the key conjecture in this work. We are expected to be convinced by the good agreement between the simulations (where the interaction are 'hand fed' into the model) and the experiments. However, the permanent quadrupoles due to contact line undulations are only expected to dominate only in the far field and require the particles to be able to reorient on the time scales of the oscillations. It should be noted that for dense layers of rough particles, rather glassy states have been reported. The near field interactions can be repulsive as well as attractive, depending on the relative orientation of the particles. This may actually be the reason no restructuring at high surface concentrations. Here lies a certain weakness of the paper, i.e. roughness should have been characterised better, or some form of non-spherical particles could have been used to impose a value of Q_2/a . Overall the estimates of the other parameters are reasonable but coarse estimates. An experiment where this roughness was varied (by changing for example particle size for a given type of particle) would have led to a more critical assessment of the mechanism.

Overall, this is a nice paper, with unique experiments which reveal potentially interesting physics. The statement "These results also suggest that extreme deformation of soft matter offers new opportunities for pattern formation and dynamic self-assembly." is a bit far fetched. It is interesting to show that the interplay between the monopole and quadrupole leads to the stringlike ordering, which is the cool bit.

Reviewer #3 (Remarks to the Author):

This is a very original paper, reporting a new unexpected phenomenon in colloidal self-assembly. Particles at an interface can interact by the local deformation of the interface. For colloidal particles, these capillary interactions are quadrupolar in nature — monopoles correspond to symmetric displacement of the interface which is normally not observed since colloids are of negligible weight. However, the submitted manuscript shows that when the interface is excited dynamically via ultrasound, the inertia of the particles starts to play a role and induces a monopolar interaction. This gives rise to new interactions, and new types of self-assembly that is evidenced by careful and elegant experiments. In addition, further credibility of the new mechanism is provided by particle simulations that are able to reproduce the phenomena, semi-quantitatively.

Given the originality and the quality of the work, and the broad general interest of the problem — soft interfaces under extreme deformations — I strongly recommend this work for publication in Nature

Communications. I do have a number of questions/comments that I trust the authors will be able to take into account.

1. Abstract: "Remarkably, we find evidence of inertial effects". Upon first reading, I was not surprised to see inertial effects at these huge accelerations — though I agree that the type of emerging interaction is truly remarkable. Perhaps this point could be phrased more concisely.

2. Fig. 1a: frankly, it took me a long time to see the change in radius. Perhaps also indicate the maximum bubble radius?

3. For readability, I suggest to motivate the scaling law for the kinetic energy (page)and also for the Weber number.

4a. The Weber number is usually associated to the kinetic energy $\rho \cdot U^2$ (or: on the force induced by the acceleration $\sim U^2/\text{length}$). Here it is not based on the kinetic energy, but on the acceleration $\Delta R \cdot \omega^2$. While I believe this is correct, this deserves to be explained — in particular since the scaling is different from the kinetic energy discussed above.

4b. I'm not a purist and I don't know the history of the Weber number, but perhaps it is not the correct nomenclature to describe unsteady effects. Similar comment for the Reynolds number that is discussed.

5. The discussion of the order parameters (defined only in the supplement) is rather vague for non-specialists. Please motivate more strongly the physical meaning of ψ_2 and ψ_3 . Can this be linked to the symmetry of the quadrupole?

6. As a general comment, in my view there is an overload of referencing to the Supplement.

7. page 7: I did not understand why $\beta = 6\pi$ was discussed for bulk particles — the entire paper is about interfaces. Is it only to connect to the classical Stokes drag? This should be mentioned. To appreciate this, it would also help to write the scaling with viscosity, to make f_{hydr} a dimensionless function.

8. Figure 3d: an inset similar to Figure 4 would be very insightful for the reader.

9. Why is the surface coverage ϕ important in the experimental regime diagram (fig 4), but not in the numerics (fig 3)?

10. Finally, could the authors motivate in the conclusion how these transient structures help towards new self-assembly applications? After all, one might argue that these transients are difficult to exploit in application.

Responses to Reviewers

Reviewer #1 (Remarks to the Author):

“The authors of this paper report fantastic experiments on the dynamical assembly of microparticles trapped at the air-water interface, i.e. on the surface of a bubble, and driven by high-frequency compression-expansion cycles. Driven by inertia, the particles, which spontaneously possess static capillary quadrupoles due to contact line pinning along an undulated line, take up additional interactions during the oscillations, i.e. capillary monopoles driven the inertial deformation of the interface and repulsive hydrodynamic forces coming from recirculation flow. The delicate balance of these dynamic interactions affects the microstructure of the interface leading to the transient formation of networks of string-like aggregates, which relax and disappear after cessation of the oscillations.”

“The experiments are highly challenging, given the extreme deformation rates, and open up a new range of phenomena that are usually inaccessible for colloidal systems. The interpretation of the data, aided by careful numerical simulations, is extremely elegant and supported by extensive sets of controls. The paper is very well written and easily accessible for the general readership of Nature Communications. I can only recommend publication.”

Authors' response: We thank the reviewer for his/her enthusiastic comments on our work, and for the positive view on the broad interest of the paper.

“Before that, I have a small set of remarks and questions, that I would like to point out to the authors.”

“1) In the videos capturing the full dynamics during the cycles, e.g. in particular Video S1, it is clearly visible that only some of the initial aggregates are rearranged and that the interparticle distance between different particles oscillates by different extents. Can the authors comment on this issue? Is it to be understood as only coming from polydispersity of the quadrupolar interactions between

particle pairs? Or is there the possibility that other elements partake the force balance? In particular, I am thinking about higher-order capillary interactions, where possible short-range repulsion can exist (e.g. as shown in the previous works of one of the authors *Soft Matter*, 2013,9, 779-786). The exploration of these effects, both polydispersity and additional interactions is highly interesting, and worth discussing.”

Authors’ response: First we comment on the issue of the interparticle distance. The distance would indeed oscillate by the same extent, as can be expected from the geometry of the problem, if the particles were uniformly distributed, and the interactions were repulsive. Because in our system the initial structure is aggregated, the particles are not free to follow the radial streamlines of the flow until all the particles are redispersed. To support this argument, we report in the histogram below a result from simulations with monodisperse particle size and quadrupole amplitude, showing the initial (yellow bars) and dynamic (blue bars) interparticle distances during oscillation. The broader distribution at maximum expansion (when the interparticle distance is also a maximum) confirms the heterogeneity of interparticle distances during expansion even in this ideal case, indicating that it is not due to polydispersity.

On the second point: we agree with the Reviewer that polydispersity in size (different monopole amplitudes) and in contact line (different quadrupole amplitudes) is an important aspect that has not been discussed in the paper. We have added a sentence Section 4 of the Supporting Information, where the model is described in detail.

SI Page 11: *“This is equivalent to assuming that the particles are monodisperse and all have the same contact angle.”*

SI Page 11: *“This quantity may also be polydisperse in experiments, due to the variability in contact line undulations between different particles.”*

With regards to higher-order capillary interactions: we cannot rule out the presence of these contributions. If near-field repulsion is present, one can expect the particles to not be at contact. Any separation distance is however below the resolution of our measurements, as the first peak of $g(r)$ measured experimentally is at $r = 2a$. The main reason we didn’t include higher-order terms was in the spirit of developing the simplest model that qualitatively captures the transient microstructure observed in experiment. We have added a sentence in the Supporting Information to motivate this assumption:

SI Page 11: *“Higher-order multipoles have been neglected for simplicity, because the experimental findings are captured to a sufficient level of detail by this minimal model.”*

This comment, together with a similar comment by Reviewer 2, also prompted us to think more about this issue. Higher-order multipoles would only have a measurable effect in the near field; the fact

that the interparticle distance periodically increases during oscillations probably enables the far-field effect of the quadrupole to dominate.

“2) The authors mention in passing (page 8) the role of the curvature of the interface, emphasizing that the effect of the dynamic monopoles is observable thanks to the spherical curvature and that a flat interface would not give the same effect. It would be useful for the reader to have more details on this, especially in the sense of relative inter-particle separation to R_0 , to identify when can the interface be assumed as flat, e.g. to guide new experiments. It is not entirely clear to me where the role of the interface curvature directly enters the various force components in Eq. (1).”

Authors' response: We have addressed this comment by moving the explanation of this subtle effect from the Supplementary Information (paragraph 4.2.4) to the main text. We have added the following paragraph:

Page 8: *“The distance between two particles positioned on a sphere having time-dependent radius $R(t) = R_0 + \Delta R \sin(\omega t)$ varies as $d(t) = d_0 + \Delta d \sin(\omega t)$. If the particles are initially in contact, the maximum separation distance is given by $\Delta d = a\Delta R/R_0$. The time average of the monopole-quadrupole force over one period of oscillations, $T = 2\pi/\omega$, is given by $\langle F_{02} \rangle_T \propto -\frac{3}{2}\gamma a^2 Q_0 Q_2 \frac{1}{d_0^3} \frac{\Delta d}{d_0}$ and is therefore non-zero only for a curved interface ($\Delta d \neq 0$).”*

We have also added the experimental range for a/R_0 :

Page 10: *“ $a / R_0 \sim 10^{-2} - 10^{-1}$ ”*

“3) Can the authors say something on the relaxation of the local direction of the quadrupoles of each particle after cessation of the oscillations? Do the particles suddenly reorient in local clusters with minimal interface deformation or is there a gradual re-orientation? This can only be accessed in the simulations, but it is interesting as a question for the design of future experiments.”

Authors' response: We thank the Referee for raising this point. We had not included results of simulations of the relaxation process, which indeed can show both the positional and orientational relaxation of the particles. We have now included this result as Section 4.9 in the Supplementary Information and a new Supplementary Figure S12:

These simulations (for $\phi = 0.4$) show that the microstructure gradually relaxes over the course of tens of ms. This additional information on the relaxation process also helped us address comment #10 by Reviewer 3 on the possibility of exploiting these transient microstructures for self-assembly.

To answer the question from the Reviewer more precisely: at the local scale of a single particle, reorientation is also not sudden. We have developed simple scaling arguments to estimate translational and rotational relaxation timescales for a particle (5-10 ms). These scalings have also been included in Section 4.9 of the Supplementary Information.

SI Page 24: “Scaling laws can be obtained for the rotational and translational relaxation times of a single particle by equating the capillary torque and force with the viscous drag. Both the rotational and translational drag are approximated with half of Stokes' drag for particles at a liquid-gas interface. The torque between two particles at contact due to quadrupolar interactions, T_{22} , balances the rotational drag $\zeta_r \approx 4 \pi \eta a^3 \dot{\phi}$, giving a characteristic rotational velocity $\dot{\phi} \approx \frac{3 Q_2^2 \gamma}{8 \eta a^3}$. The rotational relaxation time can be estimated as the time to rotate of an angle π : $\tau_r \approx \frac{8 \pi \eta a^3}{3 Q_2^2 \gamma}$. Using the typical values from the simulations for viscosity, particles radius, quadrupole amplitude and surface tension, we find a rotational relaxation time $\tau_r \approx 10$ ms. The quadrupole-quadrupole interaction force between two particles at contact balances the translational drag $\zeta_t = 3 \pi \eta a V$ to give a typical translational velocity $V \approx \frac{1 Q_2^2 \gamma}{2 \eta a^2}$. The translational relaxation time can be estimated as the time for a particle to move by a distance a : $\tau_t \approx \frac{2 \eta a^3}{Q_2^2 \gamma}$. Using the typical values for the parameters, we find a translational relaxation time, $\tau_t \approx 5$ ms”

“4) I was intrigued by the reference to shock propagation in colloidal systems, but the reference appears to be incorrect, since this work covers experiments on slow oscillatory shear.”

Authors' response: We have corrected the mistake and Ref. [25] is now the correct one: Buttinoni,

I. et al. Direct observation of impact propagation and absorption in dense colloidal monolayers. Proc. Natl. Acad. Sci. USA 114, 12150–12155 (2017).

Reviewer #2 (Remarks to the Author):

“This manuscript present some uniuqel experiments on particles on bubble interfaces and suggest a subtle interplay between capillarity and hydrodynamic disturbance velocity fields, which is termed “dynamic capillarity” by the authors. The authors study pattern formation as a function of frequency and composition and compare experiments at high frequencies with simulations.”

Authors’ response: We thank the reviewer for his/her remarks on our experiments.

“The authors demonstrate experimentally the occurrence of intriguing stringlike structures, at low to intermediate volume fractions, which naively would bring up the idea of a dominant dipolar interaction.”

“This brings me to a first question: - whereas the authors rule out all electrostatic effects - due to screening by salt in the water phase - what happens on the air side? Static friction effects have been suggested to play a role in oil water emulsions (see Scientific Reports volume 4, Article number: 4778 (2014)) - is there a possible residual charge or a static charging by friction with the air? I’m personally skeptical of the relevance of the charging reported in scientific reports, but it may be something to consider or at least comment on.”

Authors’ response: indeed we cannot rule out residual charges at the particle surface due to the friction with air. However, if these charges are present, this would not affect the qualitative picture proposed in the paper. It was our oversight to not discuss this point, which we now briefly mention in the paper. We have added this sentence:

Page 2: *“We found no evidence of additional electrostatic interactions arising from an asymmetric charge distribution on the particles at the interface (Pieranski, PRL1980)”*

To address the Referee’s comment more specifically:

- a) the paper in Scientific Reports shows that the resulting interaction energy is on the order of 10 kT, much smaller than the typical magnitude of capillary interactions (~18 kT per nm² of interface deformation for air-water).
- b) If charges were present on the air side, leading to an asymmetric charge distribution, then the particles would behave as electric dipoles perpendicular to the interface. These interactions would be repulsive and isotropic [Pieranski PRL 1980].

“The authors argue that another possibility for a directional interaction with dipole-like symmetry is that between a monopole and a quadrupole. The hypothesis that capillary deformations forces due to contact line undulations interact with dynamic monopoles seems to be the key conjecture in this work. We are expected to be convinced by the good agreement between the simulations (where the interaction are ‘hand fed’ into the model) and the experiments.”

“However, the permanent quadrupoles due to contact line undulations are only expected to dominate only in the far field and require the particles to be able to reorient on the time scales of the oscillations. It should be noted that for dense layers of rough particles, rather glassy states have been reported. The near field interactions can be repulsive as well as attractive, depending on the relative orientation of the particles. This may actually be the reason no restructuring at high surface concentrations.”

Authors' response: we thank the Reviewer for this comment. Indeed, quadrupoles are expected to dominate in the far field. The fact that the interparticle distance increases during oscillations, in particular particles are no longer in contact, plays an important role. The increase in distance makes it possible for the quadrupoles to dominate (far-field effect) and for the particles to rearrange (decrease in surface coverage at maximum expansion). We have improved Section 4.2.4 of the Supplementary Information by including a scaling argument for the timescale of local rearrangement of a particle during oscillations:

SI Page 15: “The slow dynamics of emergence of strings, over a timescale $\tau \gg 2\pi/\omega$, can be understood by balancing the mean force over one period of oscillation, $\langle F_{02} \rangle_T \propto -\frac{3}{2}\gamma a^2 Q_0 Q_2 \frac{1}{a_0^3} \frac{\Delta d}{a_0}$ with the translational drag on a particle at the interface, $\zeta_t \approx 3\pi\eta aV$, with V its characteristic velocity. Note that the drag force is approximated with half of Stokes' drag because the particle is at a liquid-gas interface. The timescale for a particle to translate over a distance equal to its radius a is given by $\tau_{t,02} \approx \frac{16\pi a^3 \eta}{\gamma Q_0 Q_2} \approx 5 \text{ ms}$. This is to be compared to the period of oscillations, which for forcing at a frequency $f = \omega / 2\pi = 40 \text{ kHz}$ is $\tau = 25 \mu\text{s}$, consistent with a slow evolution over 200 cycles.”

With regards to repulsive near-field interactions, we cannot exclude them, but we do not see an effect in our experiments, presumably due to the limited spatial resolution (pixel size of high-speed camera is $20 \mu\text{m}$ – resolution is almost 10 times lower than for normal cameras). We have clarified the assumption of neglecting higher-order multipoles on page 11 of the Supporting Information.

Here lies a certain weakness of the paper, i.e. roughness should have been characterised better, or some form of non-spherical particles could have been used to impose a value of Q_2/a . Overall the estimates of the other parameters are reasonable but coarse estimates. An experiment where this roughness was varied (by changing for example particle size for a given type of particle) would have led to a more critical assessment of the mechanism.

Authors' response: We fully agree with the Reviewer on the importance of the roughness as an experimental control parameter. We did indeed change the particle size for a given type of particle. The radius was varied from 0.9 to $5 \mu\text{m}$ in the experiments, while keeping the same surface chemistry (Latex particles, sulfate coated). Assuming that the surface roughness remains the same, this is equivalent to varying the ratio Q_2/a over one decade in the experiments. In the experiments we found no trend depending on the particle size, and therefore we presented all the results in the regime map of Figure 4.

Experiments with more controlled roughness or non-spherical particles would be interesting but more challenging. Until very recently (Zanini et al., 2017, Ref 31 in the revised version), control of the roughness of microparticles was not experimentally feasible. On the other hand, using non-spherical particles (such as ellipsoids) would lead to strong quadrupolar interactions, which would dominate over the dynamic effects in the range of parameters accessible in our experiments (oscillation amplitude and frequency). Experiments with non-spherical particles would be very interesting and will require a modified experimental design.

Overall, this is a nice paper, with unique experiments which reveal potentially interesting physics. The statement “These results also suggest that extreme deformation of soft matter offers new opportunities for pattern formation and dynamic self-assembly.” is a bit far fetched. It is interesting to show that the interplay between the monopole and quadrupole leads to the stringlike ordering, which is the cool bit.

With this sentence we have tried to open up the perspective for the broad audience of the journal, since the interplay between a static quadrupole and a dynamic monopole is a very specific observation that is likely to be of interest mostly to researchers from our community. We would like to communicate to readers outside of our immediate community the potential to use “extreme deformation” also in other systems to create new types of structures. We realised also from comment #10 by Reviewer 3 that this sentence was not sufficiently convincing. In response to that comment, we have added some information in the Conclusions on the fact that the relatively slow timescale of relaxation enables to process the microstructure, for instance by photopolymerization.

Reviewer #3 (Remarks to the Author):

“This is a very original paper, reporting a new unexpected phenomenon in colloidal self-assembly. Particles at an interface can interact by the local deformation of the interface. For colloidal particles, these capillary interactions are quadrupolar in nature — monopoles correspond to symmetric displacement of the interface which is normally not observed since colloids are of negligible weight. However, the submitted manuscript shows that when the interface is excited dynamically via ultrasound, the inertia of the particles starts to play a role and induces a monopolar interaction. This gives rise to new interactions, and new types of self-assembly that is evidenced by careful and elegant experiments. In addition, further credibility of the new mechanism is provided by particle simulations that are able to reproduce the phenomena, semi-quantitatively.

Given the originality and the quality of the work, and the broad general interest of the problem — soft interfaces under extreme deformations — I strongly recommend this work for publication in Nature Communications. I do have a number of questions/comments that I trust the authors will be able to take into account.”

Authors’ response: We thank the Reviewer for his/her interest and positive assessment of our work

“1. Abstract: “Remarkably, we find evidence of inertial effects”. Upon first reading, I was not surprised to see inertial effects at these huge accelerations — though I agree that the type of emerging interaction is truly remarkable. Perhaps this point could be phrased more concisely.”

Authors’ response: What we meant to highlight is that inertial effects *in a colloidal system* are usually not expected. We have rephrased the sentence in question to make it clearer:

Page 1: “*Remarkably for a colloidal system, we find evidence of inertial effects, caused by accelerations approaching 10,000 g*”

“2. Fig. 1a: frankly, it took me a long time to see the change in radius. Perhaps also indicate the maximum bubble radius?”

Authors’ response: The difficulty in visualising more clearly the change in radius lies in the requirement to keep the oscillation amplitude small, in order to avoid particle expulsion. Based on the advice of the Reviewer, we have added a circle indicating the maximum radius of the bubble in Figure 1(a). New figure:

“3. For readability, I suggest to motivate the scaling law for the kinetic energy (page) and also for the Weber number.”

Authors’ response: for the kinetic energy, we have now defined the velocity scale in the text:

Page 3: “*the kinetic energy of a particle in our experimental conditions, based on the maximum velocity of the interface $\omega\Delta R$* ”.

For the Weber number, please see response to the next comment.

“4a. The Weber number is usually associated to the kinetic energy ρU^2 (or: on the force induced by the acceleration $\sim U^2/\text{length}$). Here it is not based on the kinetic energy, but on the acceleration $\Delta R \cdot \omega^2$. While I believe this is correct, this deserves to be explained — in particular since the scaling is different from the kinetic energy discussed above.”

Authors’ response: because the core argument of our paper invokes the inertia of a colloidal particle, we have indeed constructed a Weber number from the ratio of inertial forces to capillary forces. The Weber number obtained from the ratio of kinetic energy to surface energy reduces to our expression using the approximation (always valid in our experiments) $a/\Delta R \sim 1$. We have added an explanation in the Supplementary Information, where the Weber number is defined:

SI Page 6: “*The Weber number based on the kinetic energy returns the same expression because in our system $a/\Delta R \sim 1$.*”

4b. I’m not a purist and I don’t know the history of the Weber number, but perhaps it is not the correct nomenclature to describe unsteady effects. Similar comment for the Reynolds number that is discussed.

Authors’ response: we have clarified in the text that the quantities of interest (Weber number, kinetic energy, etc) are based on the maximum velocity or maximum acceleration. These are usually taken to be the characteristic velocity and acceleration scales for oscillatory motion.

5. The discussion of the order parameters (defined only in the supplement) is rather vague for non-specialists. Please motivate more strongly the physical meaning of ψ_2 and ψ_3 . Can this be linked to the symmetry of the quadrupole?

Authors’ response: Similarly to the ψ_6 two-dimensional bond-orientational order parameter that measures the orientation and degree of hexagonal order around a particle, the ψ_2 function is maximised when particles form a line, and ψ_3 is maximised for a particle with neighbours respecting an sp^2 symmetry. We have clarified this in the text by modifying the following paragraph:

Page 5: “*Similarly to the ψ_6 two-dimensional bond-orientational order parameter that measures the orientation and degree of hexagonal order around a particle (6-fold symmetry),*

the ψ_2 function is a metric for the alignment of particles (2-fold symmetry). We also observe an increase of the ψ_3 order parameter, representing particles having three neighbours organized with a sp^2 (3-fold) symmetry.”

Interestingly, this cannot be linked to the symmetry of the quadrupole, because the quadrupole promotes hexagonal order. This is a signature of the monopole-quadrupole interaction.

6. As a general comment, in my view there is an overload of referencing to the Supplement.

Authors' response: we have realised this, and have reduced the number of citations from 22 to 16. We cannot further decrease the references as the amount of information included in Supporting Information requires this minimum number of references in the main text.

7. page 7: I did not understand why $\beta=6\pi$ was discussed for bulk particles — the entire paper is about interfaces. Is it only to connect to the classical Stokes drag? This should be mentioned. To appreciate this, it would also help to write the scaling with viscosity, to make f_{hydr} a dimensionless function.

Authors' response: inspired by the model by Riley (Ref. 39 in the paper) we have assumed a spatial dependence of the leading order term of the velocity field, given by $f_{\text{hydr}} = \eta\omega a^2/d^2$. We effectively account for the fact that the particles are at the interface by using a numerical pre-factor β . The case $\beta = 6\pi$ for bulk particles is discussed as a reference, since the value of β is not known for particles at an interface. We carefully evaluated the effect of β in Figure 3d and found a relatively small dependence of the final microstructure on its value. To clarify our reasoning in the text, we have modified the sentence as follows:

Page 7: “The case of a sphere in a bulk fluid corresponds to $\beta = 6\pi$, and β could be larger or smaller for particles at an interface.”

Finally, the reason why we did not make f_{hydr} dimensionless is to match the dimensions of the other functions in Equation (1), because we want to highlight the dependence on the amplitudes Q_0 and Q_2 . In double-checking all the dimensions, we have also realised that we had some typos in the expressions of f_{00} , f_{22} and f_{02} , which have now been corrected. These were only typos in the main text, and not in the Supporting Information (where the full equations are derived), or in the simulation code.

8. Figure 3d: an inset similar to Figure 4 would be very insightful for the reader.

Authors' response: we thank the Reviewer for this suggestion. We have used the pictures from Supplementary Figure S10 b, c, d as insets for Figure 3d.

9. Why is the surface coverage ϕ important in the experimental regime diagram (fig 4), but not in the numerics (fig 3)?

Authors' response: if we interpret the Reviewer's question correctly, it seems that we have created some confusion by presenting two different regime maps, which are not meant to be compared directly. The regime map in Figure 3 (numerics) presents the effect of the unknown parameters Q_0 and β , and shows that strings form for Q_0 of moderate amplitude, while the choice of β does not affect the results qualitatively. All the simulations are for the same surface coverage ($\phi = 0.4$), which was taken in the region where string formation is observed in experiment. The regime map in Figure 4 (experiments) shows the dependence on experimental parameters: surface coverage ϕ , and a measure of the dynamic effects, given by the Weber number based on the maximum acceleration.

10. Finally, could the authors motivate in the conclusion how these transient structures help towards new self-assembly applications? After all, one might argue that these transients are difficult to exploit in application.

Authors' response: we thank the Reviewer for encouraging us to discuss this point in more detail. We have now elaborated more on the point that despite being transient, the microstructure has a sufficiently long relaxation time (see also new simulation data in the response to Reviewer 1) that it can be cross-linked, for instance using polymer-functionalised microparticles and high-throughput methods like photopolymerization. We have added a reference to Dendukuri, D., Pregibon, D. C., Collins, J., Hatton, T. A. & Doyle, P. S. Continuous-flow lithography for high-throughput microparticle synthesis. *Nature Materials* 5, 365369 (2006), which is Ref. 44 in the revised version. Here, cross-linking of the entire volume of a microparticle was achieved in less than 100 ms. We have added the following text in the conclusion:

Page 11: "Despite being transient, the microstructure has a sufficiently long relaxation time that it could be exploited for subsequent processing by high-throughput methods, for instance photopolymerisation [44]."

REVIEWERS' COMMENTS:

Reviewer #1 (Remarks to the Author):

I am fully satisfied with the response to the comments of all 3 referees and with the corresponding revisions. I therefore recommend publication.

Reviewer #2 (Remarks to the Author):

The authors have addressed the comments and concerns satisfactory. I have no further comments, apart from the fact that i remain critical that this can actually be used to assemble things.

Reviewer #3 (Remarks to the Author):

I thank the authors for their detailed reply and for the various clarifications in the manuscript. I warmly recommend publication in Nature Communications.

Responses to Reviewers

Reviewer #1 (Remarks to the Author):

I am fully satisfied with the response to the comments of all 3 referees and with the corresponding revisions. I therefore recommend publication.

Authors' response: We thank the reviewer for this positive assessment and recommendation.

Reviewer #2 (Remarks to the Author):

The authors have addressed the comments and concerns satisfactory. I have no further comments, apart from the fact that i remain critical that this can actually be used to assemble things.

Authors' response: We understand the Reviewer's criticism but we trust that we have made it clear in the manuscript that while our specific system may or may not find applications in self-assembly, we suggest that extreme deformation of soft-matter in general holds promise for new self-assembly mechanisms.

Reviewer #3 (Remarks to the Author):

I thank the authors for their detailed reply and for the various clarifications in the manuscript. I warmly recommend publication in Nature Communications.

Authors' response: We thank the reviewer for this positive assessment and recommendation.